# A distributional simplicity bias
# in the learning dynamics of transformers

**Riccardo Rende, Federica Gerace, Alessandro Laio, and Sebastian Goldt**

International School for Advanced Studies
Trieste, Italy
{rrende, fgerace, laio, sgoldt}@sissa.it

## Abstract

The remarkable capability of over-parameterised neural networks to generalise effectively has been explained by invoking a "simplicity bias": neural networks prevent overfitting by initially learning simple classifiers before progressing to more complex, non-linear functions. While simplicity biases have been described theoretically and experimentally in feed-forward networks for supervised learning, the extent to which they also explain the remarkable success of transformers trained with self-supervised techniques remains unclear. In our study, we demonstrate that transformers, trained on natural language data, also display a simplicity bias. Specifically, they sequentially learn many-body interactions among input tokens, reaching a saturation point in the prediction error for low-degree interactions while continuing to learn high-degree interactions. To conduct this analysis, we develop a procedure to generate *clones* of a given natural language data set, which rigorously capture the interactions between tokens up to a specified order. This approach opens up the possibilities of studying how interactions of different orders in the data affect learning, in natural language processing and beyond.

## 1 Introduction

The success of neural networks has been explained using various "simplicity biases", which postulate that neural networks trained using stochastic gradient descent (SGD) learn increasingly complex functions of their data, going from generalised linear models to more complex, non-linear functions [1–5]. These simplicity biases have been analysed theoretically in linear networks [6–8], in non-linear shallow neural networks [1, 9] and in the NTK regime [10–12]. The viability of such a scenario has also been demonstrated experimentally, for example in image classification with convolutional neural networks [4]. More recently, a complementary *distributional simplicity bias* was demonstrated theoretically and experimentally [13, 14]: deep neural networks learn increasingly complex approximations of the distribution of their training data. A natural question to ask is then: *Does this distributional simplicity bias extend to the realm of natural language processing (NLP), where transformers are typically pre-trained via Masked Language Modeling (MLM) or Next-Token Prediction?* As a first step in this direction, Belrose *et al.* [15] recently showed that deep transformers of the Pythia suite [16] first learn the token unigram and then the bigram distribution. However, estimating the higher-order interactions is prohibitively expensive on large corpora with many tokens due to the curse of dimensionality.

In this work, we design a novel framework based on MLM to build principled approximations of a NLP data set that captures increasingly higher-order interactions between tokens. Using these approximations or "clones", we show that deep transformers trained on natural language also exhibit a simplicity bias: they sequentially learn increasingly higher-order interactions among tokens.

38th Conference on Neural Information Processing Systems (NeurIPS 2024).

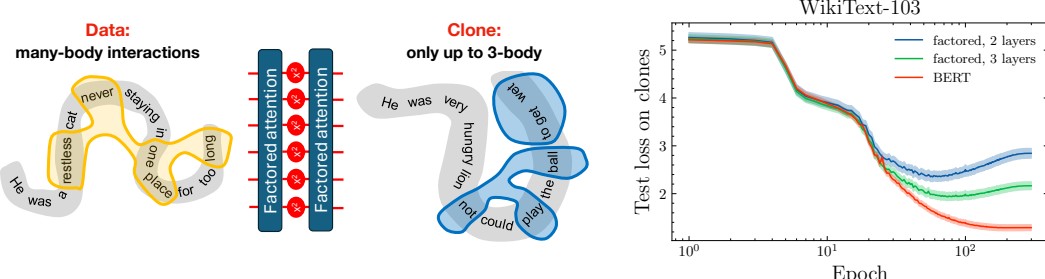

Figure 1: **Transformers learn increasingly higher-order interactions from their data. Left:** We illustrate the idea of a statistical "clone" of a data set, which approximates the underlying data distribution by keeping only interactions between tokens up to a fixed degree (in this case, three-body interactions). We introduce a principled approach to create clones by training a transformer with multiple layers of factored self-attention [17] with $x^2$ activation function between layers. The depth of the architecture controls the degree of the approximation. Clones can then be sampled from these models. **Right:** Test loss of a standard BERT-like transformer encoder [18, 19] with four attention blocks trained on the WikiText-103 [20] data set and tested on *clones* of this data set with a truncated maximum degree of many-body interactions between tokens. We show the average over five training runs starting from the same initial condition. The shaded area indicates one standard deviation.

The key idea of our framework is shown on the left of fig. 1. Given a data set containing natural language (we will use WikiText-103 [20] as a running example), we first create a number of clones. Each clone approximates the underlying data distribution of WikiText-103, but only including interactions between tokens up to a specified order (in the sketch, up to order three). We build these clones by using MLM to train a new type of transformer that consists of multiple layers of factored attention [17] with quadratic activation functions. In section 2, we show analytically that the depth of this architecture precisely controls the degree of interactions among tokens in each clone. We then sample the trained models using state-of-the-art Monte Carlo techniques [21] to obtain the cloned data sets. The clones thus form a hierarchical family of data sets with increasing order of interactions among tokens that approximate the true distribution of WikiText-103 with increasing fidelity.

Using this set of clones, we investigate the sequential learning of higher-order interactions between tokens in a standard BERT-like transformer encoder [18, 19] with four attention blocks. On the right of fig. 1, we show the test loss of this architecture during MLM *training* on WikiText-103, but *testing* on various clones of WikiText-103 with a truncated maximum degree of many-body interactions between tokens, as well as on a clone sampled directly from the trained BERT model. The test losses of the transformer on the clones are close in the beginning of training up to $\approx 20$ epochs. Beyond this threshold, BERT exhibits a different generalisation behaviour on clones that capture different orders of interaction: while the test loss on the clones containing only low-degree interactions among input tokens (blue and green curves) saturates after about 60 SGD epochs, the test loss evaluated on clones having high-order interactions (red curve) keeps improving during training. In other words, the accuracy of the transformer in the beginning of training derives from having learnt lower-order interactions, and only later during training the transformer also exploits higher-order interactions between tokens. In section 4, we extend our experimental analysis to an alternative dataset, TinyStories [22], and further include experiments involving GPT models trained on next-token prediction tasks.

The **main contributions** of this work are the following:

- We introduce generative models based on transformers with factored attention, and show that the degree of interactions they capture is rigorously controlled by their depth (section 2);
- We study sequential learning of higher-order interactions both numerically and analytically in a controlled setting where we regulate the degree of interactions between tokens (section 3);
- We train these generative models on benchmark NLP datasets, such as TinyStories [22], and sample them using Monte Carlo techniques to demonstrate the sequential learning of higher-order interactions by transformers in MLM and next-token prediction (section 4).

Our work shows that the many-body interactions between tokens in natural text are key for learning, and we note that our approach naturally extends to other data modalities.

**Further related work on simplicity biases in transformers** Bhattamishra *et al.* [23] found a simplicity bias in the ability of transformers to learn sparse boolean functions; here, we focus on the MLM task on natural language. Abbe *et al.* [24] showed that the difference between trained and initial weights progressively increases in rank throughout training, hence increasing in complexity as the task requires. Vasudeva *et al.* [25] investigated a simplicity bias in transformers in terms of their sensitivity to random changes in the input. Other recent experimental work showed that RoBERTa achieves high performance on most linguistic benchmarks early in pre-training, while more complex tasks require longer pre-training time [26], and that transformers learn different linguistic phenomena in a prototypical way [27]. To the best of our knowledge, no prior research has investigated the progressive increase in complexity of the distribution learned by a transformer from its data over time. More recently, Cagnetta *et al.* [28, 29] studied probabilistic context-free grammars and showed that increasing the training set size allows transformers to learn correlations at increasingly large distances sequentially, corresponding to their resolving increasingly higher layers of the hierarchical structure of the tree. While we were preparing the camera-ready version of this paper, Garnier-Brun *et al.* [30] demonstrated a similar phenomenon in the training dynamics of transformers.

**Reproducibility** We share the scripts to train and sample the networks and reproduce our results at https://doi.org/10.5281/zenodo.13992398.

## 2 Expressing many-body distributions with factored attention

In this section, we present generative models utilising deep networks composed of multiple attention layers. We demonstrate analytically that these models, when trained with MLM, are capable of learning increasingly higher-order interactions among tokens. In the MLM task, transformers are trained to predict missing tokens in sentences of tokenized text. We model sentences as sequences of random variables $(s_1, \ldots, s_L)$, where each token $s_i$ is an element taking values in a discrete vocabulary: $s_i \in \{1, \ldots, |\mathbb{V}|\}$. When masking the token in position $i$, the sequence $(s_1, \ldots, s_i = 0, \ldots, s_L)$ is given as input to the transformer, which embeds the input sequence into a collection of vectors $(x_1, \ldots, x_L)$, each of them living in a $d$-dimensional space. Then, this sequence of vectors is processed by the self-attention layers and transformed into a sequence of context-dependent vectors $(y_1, \ldots, y_L)$, where $y_j \in \mathbb{R}^d$ for all $j$. At the end, the output $y_i$, corresponding to the masked input token, is mapped into a vector of the same size of the vocabulary: $h_i \in \mathbb{R}^{|\mathbb{V}|}$. The prediction $p_{\mathrm{mlm}}(s_i|s_{\neq i}) = \mathrm{softmax}(h_i)$ can be finally interpreted as the probability distribution over all possible values the masked word can assume within the vocabulary. The transformer is trained to minimise the MLM cross-entropy loss, $\mathcal{L} = -\sum_{m=1}^{M} \sum_{i\alpha} s_{i\alpha}^{(m)} \log\left(p_{\mathrm{mlm}}(s_{i\alpha}^{(m)}|s_{j\neq i}^{(m)})\right)$, where $M$ denotes the total number of examples in the dataset, and $m$ indexes individual samples. The index $i$ refers to the masked token position, while $\alpha$ represents the feature index. During training, the spins are cyclically masked one by one in each sequence.

The key ingredient to the transformer architecture [31] is the self-attention mechanism, which recombines input tokens into output tokens by weighing them with an input-dependent attention matrix. Rende *et al.* [17] recently considered a simplified attention mechanism, the so-called **factored attention**, where the attention weights are input-independent and the input sequence is processed as: $y_{i\alpha} = \sum_{j=1}^{L} A_{ij} \sum_{\beta=1}^{|\mathbb{V}|} V_{\alpha\beta} x_{j\beta}$, where $A \in \mathbb{R}^{L \times L}$ and $V \in \mathbb{R}^{d \times d}$ are matrices of trainable parameters. In this way, the prediction of a single factored attention layer for the masked token is expressed in terms of the following conditional distribution (assuming the embedding matrix to be the identity):

$$p_{\mathrm{mlm}}(s_{i\alpha} = 1|s_{\neq i}) = \mathrm{softmax}\left(\sum_{j\neq i}^{L} A_{ij} \sum_{\beta=1}^{|\mathbb{V}|} V_{\alpha\beta} s_{j\beta}\right). \tag{1}$$

It can be shown analytically that, when training a single layer of factored attention via MLM, the model is capable of exactly inferring the two-body interactions among input tokens [17], when sampled according to the joint distribution:

$$p(s) = \frac{1}{Z} \exp\left(-\sum_{i,j} \sum_{\alpha\beta} J_{ij} U_{\alpha\beta} s_{i\alpha} s_{j\beta}\right), \tag{2}$$

where $J_{ij}$ describes the interactions among positions along the sequence, $U_{\alpha\beta}$ describes similarities between tokens and $Z$ normalises the distribution. In the following, we introduce the convention of using Latin indices to run over the length of the sequences $L$ and the Greek indices to run over the size of the vocabulary $|\mathbb{V}|$. As can be easily checked, the expression in eq. (1) exactly matches the conditionals associated with the two-body joint distribution in eq. (2). Remarkably, a single layer of standard self-attention, where the attention weights depend explicitly on the inputs, struggles to learn these same conditionals [17]. As a consequence, factored attention has been applied successfully in various areas of science, like protein structure prediction [32] or the approximation of ground states of frustrated quantum systems [33–36].

We now demonstrate how we can easily express distributions with many-body interactions by stacking layers of factored self-attention. In particular, we introduce an architecture that can learn interactions of a specified order by tuning the number of layers. To this end, we start considering the simplest case where we stack just two layers of factored attention, with $f(x) = x^2$ as non-linearity between them and skip connections, see fig. 2a) for a sketch of this architecture. By writing down the conditional distribution corresponding to this two-layer model:

$$p(s_{i\alpha} = 1|s_{j\neq i}) = \text{softmax}\left(\sum_{j\beta} A_{ij}^{(2)} V_{\alpha\beta}^{(2)} s_{j\beta} + \sum_{j\beta} A_{ij}^{(2)} V_{\alpha\beta}^{(2)} \sum_{k\gamma} A_{jk}^{(1)} V_{\beta\gamma}^{(1)} s_{k\gamma} \sum_{l\mu} A_{jl}^{(1)} V_{\beta\mu}^{(1)} s_{l\mu}\right),$$
(3)

we notice that the second layer expresses a three-body interaction, parameterised by a specific combination of the attention weights of the first and second layer through the tensor $\tilde{J}_{ikl} = \sum_j A_{ij}^{(2)} A_{jk}^{(1)} A_{jl}^{(1)}$ ($\tilde{U}$ is similarly expressed in terms of $V$). In particular, the first element of the sum in eq. (3) accounts for the contribution of the skip connections, while the second one matches the functional form of the exact conditionals of a three-body joint distribution:

$$p(s_{i\alpha} = 1|s_{j\neq i}) = \text{softmax}\left(3 \sum_{jk} \sum_{\beta\gamma} J_{ijk} U_{\alpha\beta\gamma} s_{j\beta} s_{k\gamma}\right).$$
(4)

Generalising these observations to a generic number of factored attention layers, we conclude that by stacking $n$ layers of factored attention with $x^2$ activation function we can infer interactions up to order $2^{n-1} + 1$. In this perspective, if the activation function is quadratic, the number of layers precisely controls the degree of interactions which can be inferred by a deep factored transformer.

## 3 Learning many-body interactions with factored attention

In this section we show how, in a controlled setup, the multi-layer factored attention model introduced in section 2 sequentially learns increasingly higher-order interactions among tokens during training. To this end, we perform a numerical experiment where we sample the input sequences from a joint distribution with tokens interacting only up to the 4th-order, that is:

$$p(s) = \frac{1}{Z}\exp\left(-\sum_{ijkl} \sum_{\alpha\beta\gamma\mu} J_{ijkl} U_{\alpha\beta\gamma\mu} s_{i\alpha} s_{j\beta} s_{k\gamma} s_{l\mu}\right).$$
(5)

Here, we ensure the couplings to be symmetric tensors by considering a set of vectors $\{j^p\}_{p=1}^P$ and setting $J_{ijkl} = \sum_{p=1}^P j_i^p j_j^p j_k^p j_l^p$. We choose $P = 200$ and we sample $j_i^p \overset{\text{iid}}{\sim}$ Bernoulli$(0.8)$. The same holds for $U$. The exact conditionals corresponding to this joint distribution can be computed analytically, precisely as in the three-body case exemplified in eq. 4. From the practical point of view, we use these conditionals to sample a set of $M$ sequences $\{s^\mu\}_{\mu=1}^M$ via Gibbs sampling, which will constitute the training set. We then train via MLM an increasing number of factored self-attention layers (from one to three) on these data, monitoring the test loss as a function of the number of epochs. The outcome of this experiment is shown in fig. 2b). The dashed orange line corresponds to the lowest possible value of the test loss that a learning model would achieve in case it would be able to perfectly reconstruct the four-body tensors $J$ and $U$ of the generative model. The dashed blue and green lines correspond instead to the lowest test loss achievable by the best two and three body approximations of the input distribution, respectively. These last two bounds can be obtained by training via MLM a model whose predictions correspond to the exact conditional of a two and

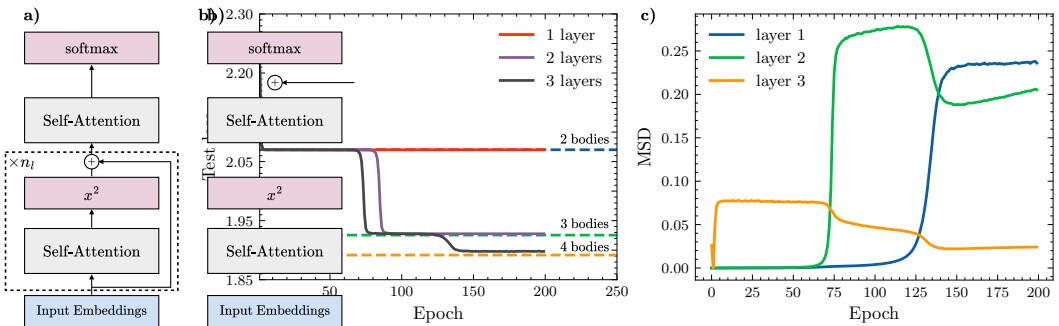

Figure 2: **a)** Multi-layer factored self-attention architecture with $x^2$ activation function. **b)** Test loss learning curves of one, two and three factored self-attention layers with $x^2$ activation function. The models were trained on a synthetic data set generated from a four-body Hamiltonian. The dashed horizontal lines correspond to the convergence value of the loss for two, three and four bodies energy based models trained on the same data set. **c)** Mean Square Displacement of the weights across different layers in a three-layers factored attention architecture. In these experiments, the size of the vocabulary was set to $|\mathbb{V}| = 10$ and the sequence length to $L = 20$. We used a training set of $M = 25600$ samples, training the models with SGD, choosing a mini-batch size of 256. The initial learning rate is chosen to be $0.1$.

three-body distribution, where the learnable parameters are the tensors in eq. (1) and eq. (4). As shown by the red curve in fig. 2b), if we train a single layer of factored attention on the four-body interacting sequences, the best we can do is to exactly recover the best two-body approximation [17]. By adding one extra layer of factored attention with $x^2$ non-linearity, we can reconstruct the best three-body approximation (violet curve). Then, the reconstruction of the full four-body interactions can be achieved by adding one further layer of factored attention (grey curve). This is consistent with the observation that by stacking $n$ layers of factored attention with $x^2$ activation function we can infer interactions up to order $2^{n-1} + 1$.

Figure 2b) also reveals that, in this controlled setup, the learning of higher-order interactions is preceded by distinct plateaus in the test loss. For example, in the three layers case, an initial plateau aligns with the best two-body approximation, followed by a subsequent plateau corresponding to the best three-body approximation. Interestingly, this sequential learning behaviour further implies that the layers are activated sequentially in time. This is illustrated in Figure 2c), where we display the Mean Square Displacement (MSD) of the weights for three layers of factored attention across training epochs, that is:

$$\text{MSD}_l(t) = \frac{1}{L^2}||A^{(l)}(t) - A^{(l)}(0)||^2 + \frac{1}{|\mathbb{V}|^2}||V^{(l)}(t) - V^{(l)}(0)||^2 \tag{6}$$

with $A^{(l)}(t)$ and $V^{(l)}(t)$ being the attention and the value matrix of the $l$-th layer at time $t$. As we can see, despite the simultaneous update of all network parameters during training, a distinctive sequential learning pattern emerges. In the early epochs, only the last layer (orange curve) activates concurrently to the appearance of the first plateau, thus suggesting that the last layer is the one responsible for learning the best two-body fit of the training data. Subsequently, the weights of the second layer come into play (green curve), and combined with the weights of the last layer can reconstruct the best three-body approximation of the underlying four-body data distribution. Close to the end of the training, also the weights of the first layer start changing and, combined with the other layers, effectively infer the ground-truth four-body interaction among tokens in the training data. A hint of the reason why the weights of the last layer find the best two-body interaction is that, due to the presence of skip connections, they appear linearly combined with the input tokens in the output prediction. In the next section we provide a more formal analysis of this phenomenon.

### 3.1 Analysing the gradient flow for two-layer factored attention

To gain theoretical insights on the sequential learning behaviour observed in the previous numerical experiments, we analyse the gradient flow of two-layers of factored attention with $x^2$ activation

function when trained via MLM. For the sake of simplicity, we restrict to the case $d = 1$, that is when the tokens in the sequence $(x_0, x_1, \ldots, x_L)$ are real numbers rather than vectors. We consider the learning framework where the model is trained to always predict the first element of each sequence, using the square loss $\mathcal{L} = \frac{1}{2}(x_0 - y_0)^2$. Here, $y_0$ represents the transformer prediction about the masked token which, in this setting, acquires the following functional form:

$$y_0 = \sum_j A_{0j}^{(2)} x_j + \sum_j \sum_k \sum_l A_{0j}^{(2)} A_{jk}^{(1)} A_{jl}^{(1)} x_k x_l . \tag{7}$$

The gradient flow equations can then be written explicitly for the attention weights of both factored attention layers as:

$$\dot{A}_{0j}^{(2)} = \left( x_0 - \sum_j A_{0j}^{(2)} x_j - \sum_j A_{0j}^{(2)} \lambda_j^2 \right) (x_j + \lambda_j^2) \tag{8}$$

$$\dot{A}_{kl}^{(1)} = 2 \left( x_0 - \sum_j A_{0j}^{(2)} x_j - \sum_j A_{0j}^{(2)} \lambda_j^2 \right) \left( A_{0k}^{(2)} x_l \lambda_k \right) , \tag{9}$$

where $\lambda_k = \sum_l A_{kl}^{(1)} x_l$. From these equations, we can immediately see that, if we initialise the network with zero weights $A^{(1)} = A^{(2)} = 0$, the first layer attention weights do not change in time, $\dot{A}_{kl}^{(1)} = 0$, while the second-layer weights do, $\dot{A}_{0j}^{(2)} \neq 0$, consistently with what we observe in fig. 2c).

We can gain some further intuition into which degree of interactions each layer learns by simplifying the data model further. In particular, we can restrict to a Gaussian design where tokens are now drawn from a zero-centered Gaussian distribution. By taking the expectation over the data, the correlations of an odd number of random variables vanish and the gradient flow equations can be written as:

$$\dot{A}_{0j}^{(2)} = h_j - \left( C A_0^{(2)} \right)_j - \sum_k A_{0k}^{(2)} \mathbb{E} \left[ \lambda_j^2 \lambda_k^2 \right] \tag{10}$$

$$\dot{A}_{kl}^{(1)} = -2 A_{0k}^{(2)} \sum_j A_{0j}^{(2)} \mathbb{E} \left[ \lambda_j^2 x_l \lambda_k \right] , \tag{11}$$

where we have defined $h_j = \mathbb{E}[x_0 x_j]$ for $j > 0$ and $C_{ij} = \mathbb{E}[x_i x_j]$ for $i, j > 0$. In appendix B we further show that, if data are Gaussian distributed, the expectation value of the loss is bounded from below by the value it takes for $A^{(1)} = 0$. Therefore, if one adds an $\ell_2$ regularisation of arbitrarily small strength, $A^{(1)} = 0$ is the only stable fixed point of the gradient flow equations. This is consistent with the numerical experiments, which show that for two-body distributed data the weights of the first layer vanish during the optimisation. In this case, only the second layer weights contribute to the learning dynamics:

$$\dot{A}_{0j}^{(2)} = h_j - \left( C A_0^{(2)} \right)_j , \tag{12}$$

whose solution describes an exponential convergence in time to $A_{0j}^{(2)}(t \to \infty) = (C^{-1} h)_j$. This stationary point reproduces the exact conditional distribution of a Gaussian entry given all the others. Indeed, considering a multivariate Gaussian distribution with zero mean, indicating the first row of the covariance matrix (except the $0, 0$ element) by $h$ and the block obtained with $i, j > 0$ by $C$ we have: $p(x_0 | x_{>0}) = \mathcal{N}(x_{>0}^T C^{-1} h, \Sigma)$. This means that the second layer is learning the covariance matrix, and thus the interactions up to the second order.

If data contains higher-order interactions among tokens, we need also the first layer to unveil the underlying data distributions. Once again, we can see this from the gradient flow equations. In the general case of higher-order interactions (thus for non-Gaussian data distributions), the stationary point $A(t) = 0$ is generally unstable. For example, considering the case of a diagonal perturbation $A_{kh}^{(1)} = \epsilon \delta_{kh}$ around $A^{(1)} = 0$, eq. (9) becomes (neglecting terms of order $\epsilon^2$):

$$\dot{\epsilon} \delta_{kl} = \epsilon A_{0k}^{(2)} \left( \Gamma_{0lk} - \sum_j A_{0j}^{(2)} \Gamma_{jlk} \right) , \tag{13}$$

where we have taken the expectation with respect to a generic data distribution and defined $\Gamma_{ijk} = \mathbb{E}[x_i x_j x_k]$. In this case $A^{(1)}(t) = 0$ is a stable solution with respect to a diagonal perturbation only

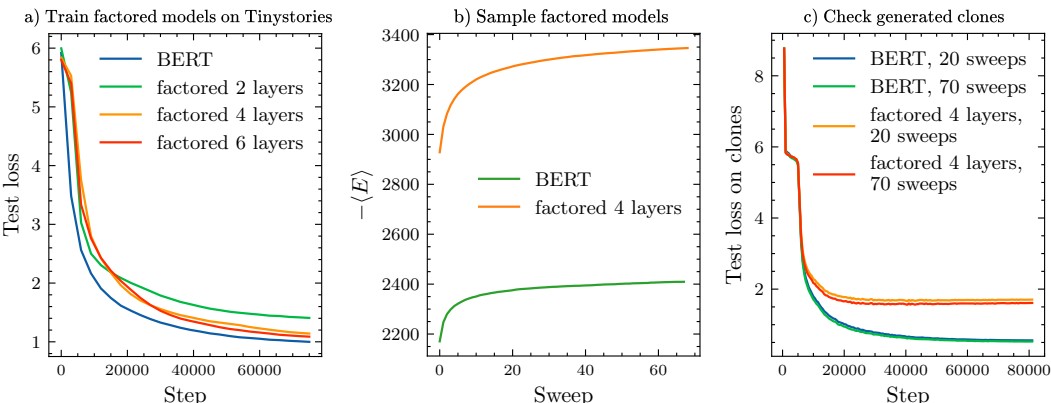

Figure 3: **Three steps for cloning a data set using factored-attention based generative models. a) Train factored-attention models on TinyStories.** Test loss curves of different factored-attention based architectures trained on TinyStories and tested on TinyStories. Specifically, we consider architectures with two, four and six factored self-attention layers with $x^2$ activation function. For comparison, also the test loss of a four-layers BERT is shown. **b) Sample factored models.** Mean score of a batch of sentences taken from the test set of the TinyStories data set and evolved with the Metropolis-Hasting sampling scheme described in appendix A.3. **c) Check generated clones.** Test loss curves of a standard four layers transformer encoder, trained on TinyStories and tested on clones generated after 20 and 70 Metropolis-Hasting sweeps. The clones were generated from a four layers standard BERT and from an architecture with four layers of factored self-attention and $x^2$ activation function (associated with a nine bodies approximation of TinyStories).

if the matrix $G_{lk} = A_{0k}^{(2)} \left( \Gamma_{0lk} - \sum_j A_{0j}^{(2)} \Gamma_{jlk} \right)$ is negative definite (implying that the perturbation decays to zero as $t$ goes to infinity). Therefore, the structure of the three-body correlations $\Gamma_{ijk}$ determines the stability of the stationary point $A^{(1)} = 0$. Indeed, if, for example, one takes $\Gamma_{jlk} = \gamma \delta_{ij} \delta_{jk}$, we have $G_{lk} = \gamma \delta_{lk} A_{0k}^{(2)} \left( \delta_{0k} - A_{0k}^{(2)} \right)$, which is not negative definite for a generic choice of $A_0^{(2)}$. This analysis allows concluding that there are conditions for which the solution $A^{(1)} = 0$ is unstable when data contains interactions of order higher than second, which is precisely in line with what we observe in fig. 2b).

## 4 Sequential learning in NLP

In this section we use the tools introduced in the previous sections to investigate which features of the input data influence the training dynamics of transformers pretrained on real NLP datasets. Specifically, we analyze the TinyStories dataset [22], training a standard BERT encoder [18] using the masked language modeling objective, as well as a standard GPT model for the next-token prediction task. Our focus lies on understanding whether a sequential learning of many-body interactions among input tokens, akin to the observed phenomenon in the synthetic scenario, is also prevalent in a real-world context. Since we do not know the underlying joint distribution over words, our approach will be to first train the attention-based models of section 2 with varying number of layers on TinyStories, that can be sampled to generate "clones" of the original data with a truncated maximum degree of interaction. We now discuss our experimental methods and analyse the results of our experiments; for full details on the methods, see appendix A.

**The clones** We exploit the generative models described in section 2 to generate clones of the original TinyStories data set (see appendix A.1 for details on how we prepare the data set). The clones are meant to approximate the underlying TinyStories distribution with increasing fidelity. We use $n = 2, 4$ and 6 layers of factored attention to generate clones that include effective interactions up to degree 3, 9, and 33, respectively. Since we use a vocabulary of size $|\mathbb{V}| = 10000$, we perform a *finite*-rank approximation of the value matrix with rank 400 (see appendix A.2 for more details). In

| | |
|---|---|
| Original text | Once upon a time there lived a cat named Max. He was a restless cat, never staying in one place for too long. One day, Max demanded something funny. He wanted a friend. |
| Factored 2 layers | Once upon a time there was a boy named Timmy. He was very hungry lion the with not could play the ball to get wet. One day, Lily saw something else. He shook his head. |
| Factored 4 layers | Once upon a time there was a boy named Tom. He saw a small cat, could not play in the park with flowers pretty. One day, Lily smelled and nice. They wanted the ball. |
| BERT | Once upon a time there was a puppy named Max. Max was a small puppy who was buzzing in a basket for very long. One day, Max saw something amazing. He wanted a puppy. |

Table 1: **Sampling the clones.** In the first row, we show part of a sentence taken from the test set of TinyStories. The second, third and fourth rows show how the original text is modified after 20 sweeps of Monte Carlo sampling associated to two and four layers factored architectures and BERT architectures, respectively.

fig. 3a), we show the test loss curves when training models with two, four and six layers of factored attention featuring the $x^2$ activation function on TinyStories. Notably, the final value of the test loss decreases systematically with the order of the interaction modelled by the architecture; for comparison, the test loss curve associated to the training of a *standard* BERT architecture [18] is also shown.

**Sampling procedure**   The bidirectional architectures we consider here are more expressive, yet also more challenging to sample than auto-regressive models such as GPT [37], as we discuss with an example in appendix A.3. Here, we adopt the sampling approach of Goyal *et al.* [21] which is based on the Metropolis-Hastings (MH) algorithm. We first select 620 examples of the test set at random. For each input sequence $(s_1, \ldots, s_L)$, we define its score in terms of the masked conditionals used for training the masked language models, $E(s_1, \ldots, s_L) = -\sum_{i=1}^{L} h_i \cdot s_i$, with $h_i$ being the logits as defined in section 2. Then, a MH sampler is run on the joint distribution defined as $p(s_1, \ldots, s_L) \propto \exp\left[-E(s_1, \ldots, s_L)\right]$. Starting from an initial sequence $(s_1^{t=0}, \ldots, s_L^{t=0})$, at each step a local move is proposed, in which the $i$-th token is masked and replaced with $s_i^{t+1} \sim p_{\text{mlm}}(s_i | s_{\neq i}^t)$. The move is then accepted according to the standard MH criterion. In this framework, the conditional distributions obtained after training the transformer with MLM are used as the transition kernel of the MH algorithm. In fig. 3b), we show the evolution of the mean score after each sweep of the sampling procedure, in the case of the standard BERT transformer and the four layers factored architecture. In table 1, we finally show how one randomly selected part of a story from the test set of TinyStories is modified after 20 sweeps of Monte Carlo sampling associated to two and four layers of factored attention, and four layers BERT architecture.

**Validating the clones**   As a sanity check of the statistical consistency of the clones, we trained a standard four-layer transformer encoder on TinyStories, but tested it on the clones generated after 20 and 70 MH sweeps, see panel c) of fig. 3. For both the clones generated from a four-layer factored architecture and from a standard four-layer transformer encoder, we can check that the test loss after 20 and 70 sweeps is really similar, thus we decided to consider the samples obtained after 20 sweeps reliable clones of the original data set.

**Sequential learning of many-body interactions by BERT**   As already anticipated, we use these clones to investigate how a standard transformer encoder trained on MLM learns the different degrees of interactions among input words. Our results in fig. 1 already suggested a sequential learning of the various degrees of interaction among words over time for the WikiText-103 dataset. In fig. 4, we show an additional experiment where we trained the same transformer (for more details about the architecture see appendix A.4) on a different data set, Tinystories [22], finding the same sequential learning dynamics. We show this effect in the left panel of fig. 4, where we plot the test loss of the four-layer BERT model of fig. 1 *trained* on TinyStories, while *tested* on clones with a fixed maximum

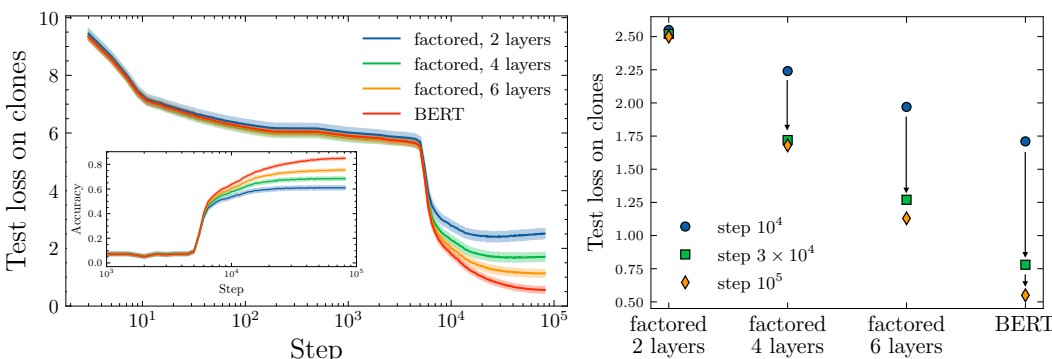

Figure 4: **BERT models trained on masked-language modelling learn increasingly higher-order interactions during training. Left panel:** In an experiment analogous to the one shown in fig. 1, we show the test loss of a standard BERT-like transformer encoder trained trained on the TinyStories data set [22] and tested on *clones* of this data set with a truncated maximum degree of many-body interactions between tokens. The inset shows the corresponding test accuracy. We show the average over five different training runs, all starting from the same initial condition. The shaded area indicates one standard deviation. **Right panel:** An alternative way to visualise the data from the left panel is to plot the test loss at steps $10^4$, $3 \times 10^4$, and $10^5$ (blue, green and orange points respectively). This visualisation highlights the sequential learning of higher-order interactions, showing that for the clones derived from two- and four-layer factored architectures the loss saturates after $3 \times 10^4$ training steps, while on the clones derived from a six-layer architecture, as well as for the clone sampled from a BERT model, the test loss continues to decrease, as indicated by the black arrows.

degree of interactions that we generated. After the initial plateau, which finishes at $\sim 5000$ training steps, the loss curves on the various clones continue to overlap up to $\sim 6000$ steps, despite the quick improvement in test error. This behaviour suggests that in the early stages of training, the model primarily learns low-order interactions without yet capturing high-order ones. Furthermore, in the final phase of training, the test error on the three-body clone, derived from a two-layer factored architecture, shows little change after $3 \times 10^4$ steps (blue curve). A similar behaviour happens for the clone obtained from a four-layer architecture (green curve). Instead, on the clone obtained from a six-layer factored architecture and from a BERT architecture, the loss continues to decrease (orange and red curves respectively) until the end of the training process, highlighting the sequential learning of the higher-order interactions by BERT. In the right panel of fig. 4, we present an alternative visualization of the same test loss values on the clones at steps $10^4$, $3 \times 10^4$, and $10^5$ (blue, green and orange points respectively). This visualization highlights that, for the clone derived from a two-layer factored architecture, training beyond $10^4$ steps does not change the test error significantly. In contrast, for the clone based on a four-layer factored architecture, the test loss shows significant improvement between $10^4$ and $3 \times 10^4$ steps, but effectively saturates after $3 \times 10^4$ steps. Notably, for the clone derived from a six-layer architecture, as well as for the clone sampled from a BERT model, the test loss continues to decrease throughout the entire training process, as indicated by the black arrows.

**Sequential learning of many-body interactions by GPT**    Finally, we extend the analysis to the learning dynamics of an autoregressive transformer model trained for the next-token prediction task. Specifically, we examine a two-layer GPT-Neo model, as also investigated in the TinyStories paper [22], training it on the original dataset and evaluating its performance on the different clones (see appendix A.4 for full experimental details). While the loss curve shows a single decay of the loss, the sequential learning is again clearly apparent, as shown in the left panel of fig. 5. An alternative visualization of the same data presented in the left panel of the same figure reveals that the test loss on clones with low-order interactions saturates within the first 500 training steps, while the test loss on clones describing higher-order interactions continues to decrease throughout the entire training period. Crucially, these findings demonstrate the applicability of our framework also to autoregressive models, which are the foundation of state-of-the-art large language models.

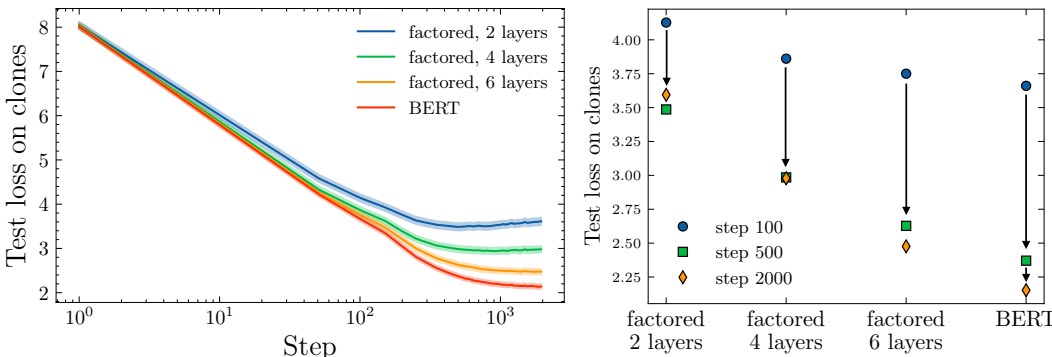

Figure 5: **GPT models trained on next token prediction tasks learn increasingly higher-order interactions during training. Left panel:** Test loss of a two-layer GPT-Neo model from [22] trained on TinyStories set and tested on clones of this data set with a truncated maximum degree of many-body interactions between tokens. **Right panel:** We repeat the analysis of fig. 4 for the GPT-Neo model trained using next-token prediction, showing again clear signatures of sequential learning. The results are consistent with respect to different random seeds.

## 5 Limitations

The biggest limitation of our methodology is associated to the sampling procedure we used to generate clones. Although we used state-of-the-art methods to sample bidirectional models [21], the presence of residual correlations cannot be excluded, even if we carefully tested the convergence of the results shown in the figures. Improving current sampling methods is an important objective for future work, but it is beyond the goals of this paper. On the theoretical side, the analytical computations of section 3 assume the embedding dimension to be $d = 1$, which is a simplification, and they do not give a quantitative estimate of the timescales over which the interactions of different orders are learnt in the numerical experiments of fig. 2, or the number of samples ("sample complexities") required to learn them. Estimating such sample complexities is an active research topic even in the case of vanilla single- and two-layer neural networks [38–43, 14]; adapting these techniques to attention-based neural networks is an intriguing challenge for future work.

## 6 Concluding perspectives

We have shown that transformer architectures trained on masked language modeling and next-token prediction tasks are able to learn the many-body interactions that govern natural text in a sequential way, going from lower to higher orders as training progresses. We showed this using a new approach to assess simplicity biases by first generating clones of a NLP data set, where the maximum level of interaction among tokens is limited to a specific order. Testing a transformer on these clones while training it on the original data set provides insights into the sequential learning patterns of the transformer. We conducted numerical simulations to show that, when training a transformer on the WikiText-103 and TinyStories data sets, the test loss curves for clones with low-degree interactions (e.g., three-body) exhibit plateaus after a fraction of the total training steps. In contrast, higher-order interactions exhibit continuous test loss reduction throughout the entire training procedure. The key element for generating the clones is the use of architectures combining factored attention layers [17] with the square activation function; the depth of these architectures controls the degree of the many-body interactions captured. In the future, it will be intriguing to extend our approach for constructing clones and for analysing the learning dynamics of transformers to different data modalities, like protein sequences [44, 32, 45]. On the theoretical side, developing a quantitative treatment of the sequential learning dynamics of (deep) transformers looms as an intriguing challenge.

**Acknowledgements**   SG acknowledges co-funding from the European Union - NextGenerationEU, in the context of the National Recovery and Resilience Plan, Investment PE1 – Project FAIR "Future Artificial Intelligence Research", and in the framework of the PRIN Project SELFMADE (code 2022E3WYTY – CUP G53D23000780001).

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

# A  Experimental details

## A.1  The data sets

**TinyStories** is a synthetic data set available on Huggingface [46] and generated by $\mathrm{GPT}-3.5$ and $\mathrm{GPT}-4$, which contains a collection of stories that typical 3 to 4-year-olds usually understand [22]. This data set was recently introduced to facilitate the development, analysis and research of LLMs. In particular, it can be used to train and evaluate LLMs much smaller than state-of-the-art models, yet still being able to produce fluent stories with consistent meaning, thus demonstrating reasoning capabilities. This has been shown even for models with one layer and below $10$ million total parameters [22]. In the same spirit of Ref. [19], we use the *full-sentences* training scheme for MLM settings. Therefore, we first concatenate all the stories of the data set, tokenising the resulting text with the $\mathrm{ByteLevelBPETokenizer}$ (as implemented in Huggingface) and setting the vocabulary size to $|\mathbb{V}| = 10000$. We then extract sequences of tokens of length $L = 128$ [47]; notice that, in the full-sentences scheme, the sequences may contain sentences that are unfinished or that cross the boundaries of different stories. In the training phase, we adopt a strategy wherein, for each sequence, we randomly mask $15\%$ of the tokens individually, as opposed to a simultaneous masking approach.

**WikiText**  We also used the WikiText-103 data set [20], which is another standard NLP data set. Similar to tinystories, we tokenized using the ByteLevelBPETokenizer, setting the vocabulary size to $|\mathbb{V}| = 500$ and considering sequences of length $L = 64$ tokens.

## A.2  Obtaining the clones

We exploit the generative model described in section 2 to generate clones of the original TinyStories data set (see appendix A.1 for details on how we prepare the data set). The clones are meant to approximate the underlying TinyStories distribution with increasing fidelity. In particular, as already discussed, by stacking $n$ layers of factored attention, we can generate copies of TinyStories whose maximum degree of interaction is $n + 1$. To enhance training stability, we normalise the activations through Layer Normalisation. Each layer of factored attention involves two matrices of parameters, $A \in R^{L \times L}$ and $V \in R^{|\mathbb{V}| \times |\mathbb{V}|}$. In our setup, the size of the vocabulary is $|\mathbb{V}| = 10000$, thus the matrix $V$ would involve 100 millions of parameters. To reduce this number, we perform a *finite*-rank approximation of the value matrix, writing $V = AB$, with $A \in R^{|\mathbb{V}| \times r}$ and $B \in R^{r \times |\mathbb{V}|}$. In practice, we choose $r = 400$. In this setting, we train two, four and six layers of factored attention via MLM. These models can then be sampled to generate three distinct clones of the original TinyStories data set, with a different degree of interaction among tokens.

## A.3  Sampling procedure

Auto-regressive architectures [37] are trained to learn the set of distributions $\{p(s_i|s_{<i})\}_{i=1}^{L}$ and can be efficiently sampled. These models are associated to a joint over the tokens through the chain rule. On the other hand, bidirectional architectures model are trained to learn the set of conditionals $\{p(s_i|s_{\neq i})\}_{i=1}^{L}$. One may be tempted to sample these models with a Gibbs sampler, however this can result in an *inconsistent* sampler, where the conditionals may not be associated to any valid joint distribution. To understand why this could be the case, consider two discrete random variables $X$ and $Y$, taking respectively $m$ and $n$ different possible values. The conditional distributions are defined in terms of two matrices $A_{ij} = P(X = x_i|Y = y_j)$ and $B_{ij} = P(Y = y_j|X = x_i)$, subject to the constraints $\sum_i A_{ij} = 1$ for every $j$ and $\sum_j B_{ij} = 1$ for every $i$. The conditionals $A$ and $B$ are compatible if and only if two vectors $\tau_i = P(X = x_i)$ and $\eta_j = P(Y = y_j)$, defining the marginals, exist such that $\tau_i B_{ij} = \eta_j A_{ij}$ for every $i$ and $j$. The latter is a linear system, which does not always have a solution (see Ref. [48] for more details and for the generalisation of the problem to more than two random variables). A possible solution is proposed in Ref. [21], where the authors develop a sampling scheme based on the Metropolis Hastings (MH) algorithm, starting from the evidence that BERT can be used to reliably score a set of sequences [49]. In this approach, for each given input sequence $(s_1, \ldots, s_L)$, its score is defined as $E(s_1, \ldots, s_L) = -\sum_{i=1}^{L} h_i \cdot s_i$, with $h_i$ being the logits as defined in section 2. Then, the MH sampler is run on the joint distribution defined as $p(s_1, \ldots, s_L) \propto \exp(-E(s_1, \ldots, s_L))$. Starting from an initial sequence $(s_1^{t=0}, \ldots, s_L^{t=0})$, at each step a local move is proposed, in which the $i$-th token is masked and replaced with $s_i^{t+1} \sim$

$p_{\mathrm{mlm}}(s_i|s^t_{\neq i})$. The move is then accepted according to the standard MH criterion. Hence, in this framework, the conditional distributions obtained after training the transformer with MLM are used as the transition kernel of the MH algorithm. In order to generate the clones of TinyStories, we choose randomly 620 examples of the test set and evolve them for 20 MH sweeps (2560 steps).

### A.4 Architecture and the training procedure

**BERT**  We use a *standard* BERT-like transformer encoder [18, 19] with input-dependent attention weights (thus using queries and keys) and MLP after each self-attention layer. We use skip connections and put LayerNorm inside each residual block, thus adopting the successful Pre-LN scheme introduced in Ref. [50]. We choose the number of layers equal to 4, number of heads equal to 16, embedding dimension equal to 768 and size of the hidden layer of the MLPs equal to 1536. Similar hyper-parameters are also used for the models trained in Ref. [22]. We choose SGD for the optimiser, setting the batch size to 1024. We start with a learning rate of 0.03, annealing it with a cosine decay scheduler. We trained the model for 10 hours on eight A100 GPUs.

**GPT**  For our experiments with next-token prediction, we trained a two-layer GPT-Neo model, as also investigated in the TinyStories paper [22]. We set the hyperparameters of the network to be: number of layers equal to 2, number of heads equal to 16 and embedding dimension equal to 768. We used the AdamW optimiser, with the same hyperparameters used in Ref. [22], namely learning rate equal to $5e - 4$, exponential decay $\beta_1 = 0.9$ and $\beta_2 = 0.95$. The weight decay was set to 0.1 and we used a batch size equal to 128.

## B  Analytical details

**Solving for $A^{(2)}$ in the case $A^{(1)}(t) = 0$**

We aim to solve the dynamical equations presented in Eqs. 9 of the main text, specifically in the scenario where the first layer is skipped, which corresponds to setting $A^{(1)}(t) = 0$. Under this condition, the dynamics of the second layer are described by:

$$\dot{A}^{(2)}_{0j} = h_j - \left( CA^{(2)}_0 \right)_j .$$

To solve this differential equation, we first write $C = U^T D U$ with $D$ a diagonal matrix with diagonal elements $D_j$ and $UU^T = I$ since the covariance matrix is symmetric. Thus we have

$$\dot{\tilde{A}}^{(2)}_{0j} + D_j \tilde{A}^{(2)}_{0j} = \tilde{h}_j ,$$

where we have defined $\tilde{A}^{(2)}_{0j} = UA^{(2)}_{0j}$ and $\tilde{h}_j = Uh_j$. In this new basis, the equations are decoupled. With the initial conditions $\tilde{A}^{(2)}_{0j}(t = 0) = \tilde{A}^{(2)}_{0j}(0)$, the solution is:

$$\tilde{A}^{(2)}_{0j}(t) = \frac{\tilde{h}_j}{D_j} + \left( \tilde{A}^{(2)}_{0j}(0) - \frac{\tilde{h}_j}{D_j} \right) e^{-D_j t} .$$

It is easy to verify that, in the original basis, the stationary solution is $A^{(2)}_{0j}(t \to \infty) = (C^{-1}h)_j$ as discussed in the main text.

**Loss lower bound**

Consider the analytical setup presented in section 3 of the main text. The mean square error loss associated with the model in Eq. 7 is:

$$\mathcal{L}(A^{(1)}, A^{(2)}_0) = \left( x_0 - \sum_j A^{(2)}_{0j} x_j - \sum_{jkl} A^{(2)}_{0j} A^{(1)}_{jk} A^{(1)}_{jl} x_k x_l \right)^2$$

where $b_i = B_{0i}$. We want to show that, in the case of Gaussian distributed data, for any values of the parameters $b_i$ it is bounded from below by the value it has in $A^{(1)} = 0$. Taking the expectation, we get:

$$\mathcal{L}(A^{(1)} \mid A_0^{(2)}) = f(A_0^{(2)}) + \sum_{jkl}\sum_{j'k'l'} A_{0j}^{(2)} A_{jk}^{(1)} A_{jl}^{(1)} b_{j'} A_{j'k'}^{(1)} A_{j'l'}^{(1)} E\left[x_{k'} x_{l'} x_k x_l\right]$$

$$= f(A_0^{(2)}) + \sum_{jj'} b_{j'} A_{0j}^{(2)} \sum_{klk'l'} A_{jk}^{(1)} A_{jl}^{(1)} A_{j'k'}^{(1)} A_{j'l'}^{(1)} \left(C_{kl}C_{k'l'} + C_{kk'}C_{ll'} + C_{kl'}C_{k'l}\right)$$

$$= f(A_0^{(2)}) + \sum_{jj'} b_{j'} A_{0j}^{(2)} \left(\Gamma_{jj}\Gamma_{j'j'} + 2\Gamma_{jj'}^2\right) = f(A_0^{(2)}) + \left(\sum_j A_{0j}^{(2)} \Gamma_{jj}\right)^2 + 2\sum_{jj'} b_{j'} A_{0j}^{(2)} \Gamma_{jj'}^2 \,,$$

where $f(A_0^{(2)})$ is a function of $A_0^{(2)}$ alone, and we defined $\Gamma = A^{(1)} C A^{(1)}$. For any possible choice of the vector $A_0^{(2)}$, we have:

$$\mathcal{L}(A^{(1)} \mid A_0^{(2)}) \geq \mathcal{L}(A^{(1)} = 0 \mid A_0^{(2)}) = f(A_0^{(2)}) \,.$$

Indeed, $\left(\sum_j A_{0j}^{(2)} \Gamma_{jj}\right)^2 \geq 0$ for all choices of $A^{(1)}$. Moreover, $\sum_{jj'} b_{j'} A_{0j}^{(2)} \Gamma_{jj'}^2 \geq 0$, since all the matrix $\Gamma_{jj'}^2$ are non-negative, therefore all its eigenvalues are non-negative.

