# OpenReview forum: "A distributional simplicity bias in the learning dynamics of transformers"
_NeurIPS.cc/2024/Conference — NeurIPS 2024 poster_

### Official Review · Reviewer_NZGs · 2024-07-04

**Soundness:** 4
**Presentation:** 3
**Contribution:** 4
**Rating:** 6
**Confidence:** 4

**Summary:**

The paper investigates whether transformers trained on natural language data using masked language modelling (MLM) exhibit a "simplicity bias" - learning increasingly complex interactions between input tokens over the course of training. The work provides both theoretical and empirical evidence for a simplicity bias in transformer models trained on text data.​​​​​​​​​​​​​​​​

The key contributions of the paper are listed below:

1) Introduction of a novel framework to study the learning dynamics of transformers:
- They develop a method to create "clones" of natural language datasets that capture token interactions up to a specified order.
- This is done by using transformers with factored attention layers and quadratic $x^2$ activation functions.

2) Demonstration of sequential learning in transformers: By using their cloning method on the TinyStories dataset, the authors show that a BERT-like transformer sequentially learns increasingly higher-order interactions between tokens. More specifically, lower-order interactions (e.g. 3-body) are learned earlier, while higher-order interactions (e.g. 7-body) continue to be learned later in the training.

3) Theoretical analysis: the authors provide analytical insights into the learning dynamics of multi-layer factored attention models, showing how different layers contribute to learning interactions of increasing complexity.

4) Evidence on language data: the authors demonstrate that interactions between words in natural text (specifically TinyStories) are highly many-body, up to at least 7th degree.

**Strengths:**

Overall, I enjoyed reading the paper very much, and I also love the hypothesis investigated in this work. The specific aspects of this work that I love are listed below.

1) *The proposed hypothesis is of great significance*:
Beyond the significance of this simplicity bias mentioned in the paper, I want to provide the perspectives from the cognitive science community. A series of language evolution studies, e.g. [1], have shown that simplicity bias plays a key role in shaping the evolution of natural languages. For syntax (i.e. generative functions of language data), humans also have such simplicity bias. It would be highly interesting and exciting (at least for me) to see that Transformer-based language models also gradually learn increasingly complicated functions for generating language.

2) *The illustration is clear, and the structure is well-organised*:
The illustration of the method and the experiments is clear, and the structure of the paper is well-organised. Therefore, the whole idea is delivered very well.

3) *The methodology is convincing*:
The study methodology, clones of datasets and factored attention, is convincing per se, and can quite well align with the hypothesis investigated in the paper. Though there is room for this work to be improved, it could be a pivot step towards understanding the learning dynamics of Transformer-based language models.


References
[1] Smith et al. (2013). Linguistic structure is an evolutionary trade-off between simplicity and expressivity

**Weaknesses:**

1) *Uncommonly used architecture*.

As a first step towards understanding the learning dynamics of Transformer-based language models, I understand the model choice of the authors. However, this also limits the contributions and impact of this work. From my perspective, the "Transformers" in the title might better be replaced by "BERT-like Language Models", as the work investigates only a particular implementation of transformers on only language modelling. So, "learning dynamics of transformers" might be far bigger than the research scope of this work.

Moreover, for language modelling, decoder-only Transformers are wider applied these days. The $x^2$ activation function studied in this work is not common for language modelling either. These two factors further limit the contributions and impact of the work.

2) *Higher orders are not straightforward to sample.*

As mentioned by the authors in Section 5, the time complexity of the sampling algorithm for creating clones is not linear with respect to the number of samples. Thus, it would become increasingly harder to sample clones of higher-orders. This is a major bottleneck to apply the work at larger scales.

3) *The empirical evidence is not strong enough.*

To me, only one model on one specific dataset is not very convincing to empirically verify the proposed simplicity bias hypothesis. It would be much more convincing if the authors can show empirical evidence on more diverse language tasks, e.g. machine translation, and etc.

**Questions:**

1) What does the "last layer" in line 163 mean?

It's unclear to me how is the "last layer" in line 163 defined. I kind of assume that it means the (bottom) layer that's closest to the inputs, a.k.a. the secondly activated layer in the model. The authors need to clarify the meaning of it to avoid unnecessaty confusion.

**Limitations:**

Please refer to my weaknesses section.

---

> ### Author Rebuttal · Authors · 2024-08-07
>
> We want to thank you for your positive comments and your enthusiam for our work. We really appreciated the
> series of language evolution studies on simplicity biases that you suggested,
> and we decided to include them in the revised version of our manuscript.
>
> Below we reply to your questions and concerns point-by-point; we hope that these points alleviate your concerns regarding the architecture, the time complexity of sampling, and the need for additional empirical
> evidence. If so, we would appreciate it if you could reconsider your rating; if
> not, we're looking forward to discussing further during the discussion period.
>
> - *Title*: We agree with your suggestion for the title, which would be more
>   specific.
>
> - *Activation function:* In our framework, we only use the $x^2$ activation
>   function in the architectures that are subsequently used to generate the
>   clones of a real NLP dataset, in which the maximum degree of interaction among
>   tokens can be controlled by tuning the number of layers. We use these clones
>   to demonstrate the simplicity bias of a standard transformer encoder
>   (BERT-like) with *standard activation function* trained under MLM on a real
>   NLP task. Thus, the main result of our paper is obtained for a standard
>   architecture, trained with a standard task on a standard dataset.
>
> - *Decoder-only transformers* are indeed popular language models. The procedure
>   we present in this manuscript, and in particular the analytic derivations, are
>   however designed for BERT-style transformers, and validated only on such
>   models. Extending it to causal models and other NLP tasks is for sure possible
>   and interesting, but non-trivial, and we plan to investigate this topic in our future research.
>
> - *Time complexity of sampling* The reviewer is right that the sampling of the
>   clones is not trivial; it is difficult to fully decorrelate the samples, and
>   the complexity of the method we use is not linear in the number of
>   samples. However, the modified transformer architectures that model the data
>   are computationally much more easy to handle than, say, estimating the
>   corresponding $n$-grams. The clone models require only a couple of layers of
>   factored self-attention, which is easier to compute than vanilla
>   self-attention layers, and they do not need fully-connected networks
>   in-between attention layers. This makes sampling of our clones, while
>   challenging, also way cheaper than sampling the distribution generated by an
>   ordinary transformer.
>
> - *Evidence for simplicity bias* We appreciate the need for additional experimental verification,
>   and have repeated our experiments on another data set, WikiText101. As we show
>   in our plot in the additional one-side PDF, we can see sequential learning of
>   increasingly higher-order interactions also on this data set. In our
>   manuscript, we further show sequential learning in a controlled setting with
>   synthetic data (cf. Section 2) and we see the same behaviour in our analytical
>   treatment of Sec. 3.1. We hope that this combination of experimental
>   verification on tinystories, WikiText101, synthetic data, combined with our
>   analytical results, is convincing evidence for our claims.

---

### Official Review · Reviewer_ex9y · 2024-07-09

**Soundness:** 2
**Presentation:** 2
**Contribution:** 3
**Rating:** 6
**Confidence:** 3

**Summary:**

This paper demonstrates that masked language models like BERT approximate distributions with increasing complexity in terms of the number of interactions, as tracked over the course of training. They do this by approximating lower order interactions with provably limited models.

**Strengths:**

This paper has one contribution that I think is very valuable: it introduces a new family of neural architectures that have a guarantee on what order of feature interaction they are capable of representing.

Furthermore, the paper is one of a few that use an approach of approximating a true distribution using an analytically characterized language; the most common way this is deployed is by training on ngram-generated text to guarantee that it is representable by only ngrams. This is a promising approach that sits at the midpoint between purely synthetic and purely natural data, and more work in machine learning should embrace it.

This paper provides another example of how models learn increasingly complex representations over the course of training. It is a good addition to the literature, although it is not the first such paper in language modeling or NLP.

**Weaknesses:**

*Post rebuttals: The authors have expressed a willingness to rewrite the misleading framing and expand their literature review, so I have raised my score from 4 to 6.*

The paper should be updated to reflect related literature in language modeling or NLP. Here are a few other examples:
- Language models start out by behaving like ngram models in their grammatical capabilities and eventually begin to act more like humans https://aclanthology.org/2022.acl-long.568.pdf
- there are well documented dependencies including the famous checkmate in one emergence example from https://arxiv.org/pdf/2206.04615
- RoBERTa achieves high performance on most linguistic benchmarks early in pre-training, but more complex tasks require longer pre-training time. https://aclanthology.org/2021.findings-emnlp.71/
- LSTMs start by learning representations that correspond to simple groupings like part of speech and eventually learn more complex language modeling features https://aclanthology.org/N19-1329/
- Section 4 explores increasingly complex ngram learning in https://arxiv.org/pdf/2402.04362
in context learning approximates shorter ngrams before longer ngram distributions https://arxiv.org/abs/2402.11004

The core issue with this paper is that at no point does it describe a simplicity bias. While conceptual dependencies and progressively complex learned distributions are clearly related to simplicity bias, these things are not the same. A simplicity bias requires one to show that the model fails to learn something difficult because it has learned something easier, but arguably what is described here might actually be a sequence of strict dependencies between interaction levels. In order to demonstrate a simplicity bias, the authors would have to show that the model fails to learn higher order interactions because of the availability of lower order interactions, but the lower order interactions are available regardless of whether the higher ordinary interactions are present so there are no conditions to contrast in this way.

Some ways of demonstrating simplicity bias are:
- demonstrating that introducing a simple rule prevents the model from learning a more complex rule https://arxiv.org/abs/2011.09468 https://arxiv.org/abs/2006.07710
- demonstrating that suppressing a simple rule allows the model to get better at learning a more complex rule https://arxiv.org/abs/2309.07311 (specifically is focused on MLMs as well)
- Correlating the accessibility of a rule and the degree to which the model ultimately captures a different rule https://arxiv.org/abs/2006.12433
- analytic proofs of a preference towards simplicity https://arxiv.org/abs/1805.08522
It is not sufficient to simply demonstrate that the model happens to learn simple things and then more complex things later on.

I want to be clear that this paper still has value, as it demonstrates yet another way in which these models learn increasingly complex distributions, which is a different but related thread of the literature. However, the framing is misleading and I do not believe it is ready for publication without a substantial rewrite.

**Questions:**

Can you give a simple explanation of what a k-body interaction would look like? is it essentially stating something about ngram models, but not restricted to the most proximate tokens?

Can you guarantee that the difference provably limited models actually differ by nothing but their limitations? What about random variation?

**Limitations:**

nothing obvious

---

> ### Author Rebuttal · Authors · 2024-08-07
>
> We thank you for your careful reading of the manuscript, your
> detailed review, and the many references. Below we reply point-by-point; we hope that our replies alleviate your concerns. If so, we would appreciate it if you could revisit your rating; if not, we look forward to discussing further during the discussion period.
>
> - *Related works* We thank the referee for the references. Indeed, the
>   connection with n-gram models is extremely important, since our clones
>   implement n-grams using a transformer architecture. We will explicitly mention
>   this in the revised version. On the other hand, we are aware that the NLP
>   community has made enormous steps forward in understanding learning dynamics,
>   with paradigms that go well beyond n-grams. In the revised version of the
>   manuscript we will explicitly mention ref
>   https://aclanthology.org/2022.acl-long.568.pdf and
>   https://aclanthology.org/2022.acl-long.568.pdf as examples of advanced
>   quantitative analysis in this direction. We will also mention
>   https://aclanthology.org/N19-1329/ as a technique which allow probing the
>   learning dynamics in language. Finally, we discuss the relation of our work to
>   the very recent paper https://arxiv.org/pdf/2402.04362. They show that deep
>   transformers first learn the token unigram and then the bigram
>   distribution. However, estimating the higher-order interactions with the approach described in this work is prohibitively expensive on large corpora with many tokens due to the curse of
>   dimensionality. Moreover, we would like to highlight that none of these papers,
>   including arXiv:2402.04362, introduces an architecture which produces
>   distributions in which the simplicity bias can be _controlled_ by tuning
>   the interactions between tokens that are captured by the model, at an arbitrary order. This
>   analytically guaranteed control, as also recognized by the reviewer, is an
>   important and novel contribution of our work.
>
> - *The nature of simplicity bias* Thank you for sharing these references on
>   various ways in which learning simple rules might interfere with learning more
>   complex rules. These papers demonstrate "simplicity biases" by performing
>   experiments on appropriately fine-tuned training tasks, where simple rules
>   based on spurious correlations hinder the network from learning the
>   generalisable input features. These behaviours are interesting and important, but have to be induced by an external intervention (a fine-tuned task).  If trained by a standard task (say text completion), transformers seem instead  to pick up the most relevant
>   underlying rules of language rather well,  producing flawless chunks of
>   text. It is this process that we want to study through the lens of another
>   line of research. We hope that our paper convincingly shows that during a single training run, a model often
>   seems to be learning increasingly complex functions or distributions that
>   explain its training data. We discuss previous theoretical and experimental
>   work in the introduction, in particular in lines 18-26, many of which have
>   been published at NeurIPS/ICML/ICLR in the last couple of years (Refs
>   2,3,4,5,6,10,12,13). We will mention this separate line of work on simplicity
>   biases in the revised version of the manuscript.
>
> Regarding your *questions*:
>
> > Can you give a simple explanation of what a k-body interaction would look
> > like? is it essentially stating something about ngram models, but not
> > restricted to the most proximate tokens?
>
> A $k$-body interaction is any interaction between $k$ tokens anywhere along the
> sequence that can not be written as a product of lower order interactions. In
> the energy-based models we use to construct the clones, each such interaction
> corresponds to one term in the energy function involving $k$ tokens. Indeed, the tokens do not have to be proximate. N-gram
> models instead simply define the number of previous tokens on which a given
> token is conditionally dependent, i.e. how many conditions there are in $p(x_n |
> x_{n-1}, x_{n-2}, …)$.
>
> > Can you guarantee that the difference provably limited models actually differ
> > by nothing but their limitations? What about random variation?
>
> The architecture of the "limited models" that we use to obtain the clones only
> differ by their "limitations", namely by the degree of the interactions between
> tokens that they cover. After training, the values of the parameters of these
> models could have some random variations in their values due to different
> initial conditions; however, this will not change the degree of the interactions
> between tokens represented by these models.

---

> > ### Comment · Reviewer_ex9y · 2024-08-07
> > **Response**
> >
> > > The nature of simplicity bias
> >
> > I do not feel that this response addresses my criticism, which is that none of the results justify framing the paper as studying simplicity bias. Although training dynamics of increasing complexity are clearly related to simplicity bias, they are not evidence of simplicity bias and therefore the paper should not be framed around describing them as such. This criticism is fairly mild, but I consider it to be crucial: I believe that the title as written is making a claim that is unrelated to the results and not supported by evidence in the paper. I don't think that this paper should be published with a misleading title or framing, although the results themselves are interesting. This issue is easily fixed; I would easily raise my score for a version of the paper that had a different title and discussed simplicity bias as related work rather than as a description of the results here.
> >
> > > The architecture of the "limited models" that we use to obtain the clones only differ by their "limitations", namely by the degree of the interactions between tokens that they cover. After training, the values of the parameters of these models could have some random variations in their values due to different initial conditions; however, this will not change the degree of the interactions between tokens represented by these models.
> >
> > Is it empirically true that models trained under different initial conditions with the same set of limitations exhibit all of the same behavior that you measure?

---

> > > ### Author Response · Authors · 2024-08-08
> > > **Reply**
> > >
> > > Thank you for engaging with our rebuttal so quickly, we appreciate it.
> > >
> > > > Re: framing
> > >
> > > We now realise that the term simplicity bias can be misleading; since we strive for maximum clarity of our paper, we are considering changing the framing of the paper by changing its title to “A bias towards low-order interactions in the early learning dynamics of BERT-like language models”, and generally changing “simplicity bias” to the more descriptive “bias towards low-order interactions” throughout the text. Would that address your criticism?
> > >
> > > > Is it empirically true that models trained under different initial conditions with the same set of limitations exhibit all of the same behavior that you measure?
> > >
> > > Yes; in the course of our work, we found that if we generate clones from networks with the same interaction-limited architecture trained from different initial conditions, the performance of a given transformer tested on the different clones will be the same to within the variation in performance of the same transformer trained from different initial conditions.

---

> > > > ### Comment · Reviewer_ex9y · 2024-08-08
> > > > **Reframing**
> > > >
> > > > I would perhaps use a framing that's not oriented around the term bias, which carries implications around the simpler solution being an actual attractor (eg, you could use snowclone boilerplate title of "Transformers learn feature interactions of increasing complexity"). However, given that you're willing to change the misleading and confusing framing, I will be raising my score, as I think that the results and method are overall valuable.

---

> > > > > ### Author Response · Authors · 2024-08-08
> > > > >
> > > > > Thank you first of all for your continued engagement, and for increasing your score, we really appreciate it. Your final suggestion of avoiding the term bias altogether is a good one; we will change the title in that sense.
> > > > >
> > > > > All the best, The Authors

---

### Official Review · Reviewer_fVhm · 2024-07-12

**Soundness:** 3
**Presentation:** 3
**Contribution:** 3
**Rating:** 6
**Confidence:** 4

**Summary:**

The paper investigates the simplicity bias in BERT-style Transformers trained with MLM. The study reveals that these models initially learn simpler interactions between tokens and gradually learn higher-order interactions. This finding aligns with simplicity biases observed in many neural network architectures. The authors develop a method to create dataset "clones" that capture token interactions up to specified orders -- which they called many-body interactions. This allows detailed analysis of the learning dynamics in Transformers.

**Strengths:**

**Novel Insight**. The paper provides novel insights into the learning dynamics of Transformers, highlighting a simplicity bias that was previously unexplored in the context of self-supervised learning with MLM. At the same time, most previous simplicity bias papers more focus on the bias at the **beginning** or **end** of training, and this papers provides this novel insight in studying the training dynamics and the arising simplicity bias explanation during training procedures.


**Many-body interactions**. The introduction of a procedure to generate dataset clones that capture interactions up to specified orders is a significant methodological contribution, facilitating deeper analysis of model learning behaviors. The proposed method enables us to have a deeper understanding of token interaction and itself is of a high contribution.

**Weaknesses:**

**Computational overhead**. The method to create dataset clones and analyze the learning dynamics is **computationally intensive**. It will potentially limit its applicability to larger datasets or more complex models. I would like to hear the authors proposals on how to deal with this issue.

**Generality** While the study focuses on BERT-style Transformers and the TinyStories dataset, the generalizability of the findings to other Transformer architectures or more diverse NLP tasks remains to be demonstrated. One thing to note, that Transformers are also applied to vision tasks, speech tasks, etc. Would like to hear how this technique can generalize to studying transformers on non-language tasks and if the conclusion would be similar. Similarly, how this would extend to decoder-only models such as GPT, remains unknown.

**Questions:**

It's a very clearly written and presented paper, I don't have further questions to ask.

---

> ### Author Rebuttal · Authors · 2024-08-07
>
> We thank you for your detailed questions and your feedback, which led us to conduct additional experiments. Below we reply point-by-point; we hope that our replies alleviate your concerns. If so, we would appreciate if you revisit your rating; if not, we look forward to discussing further during the discussion period.
>
> - *Computational cost:* applying our method to train a many-body
>   approximation of a data set, or "clone", is _less_ expensive than training a
>   standard BERT-like model, since the architecture is much simplified. Likewise,
>   the sampling procedure that we use from Goyal et al. [ICLR '22,
>   arXiv:2106.02736] can sample the standard BERT models. Hence we are confident
>   that the method can be applied to any data set on which BERT was applied to.
>
> - *Generality:* The procedure, as presented in the manuscript, and in particular
>   the analytic derivations, are designed for BERT-style transformers, and
>   validated only on such models. Extending it to causal models and other NLP
>   tasks is for sure possible and interesting, but non-trivial, as the probability structure induced by causal training is analytically (and empirically) different. We plan to investigate these
>   topics in our future research.  To further support the empirical
>   evidence of sequential learning in BERT-like architectures trained with MLM on
>   NLP datasets, we replicated the analysis on the WikiText-2 dataset, observing
>   a similar hierarchy in the  learning process, and with a vanilla transformer on the synthetic data set we created  (see plots in the attached one-page PDF). In both cases, we see clear sequential learning by the transformers.

---

### Official Review · Reviewer_Kh1E · 2024-07-13

**Soundness:** 2
**Presentation:** 2
**Contribution:** 2
**Rating:** 5
**Confidence:** 3

**Summary:**

This paper shows that transformers sequentially learn high-order interactions between input tokens during the training process, which echoes the simplicity bias prevalent in neural networks. Specifically, the paper proposes a method to extract certain orders of interactions from the original training set, and construct a new dataset that contains purely low-order interactions. Then, the new dataset is used to evaluate how well a transformer learns certain orders of interactions.

**Strengths:**

The paper attempts to validate the simplicity bias on neural networks trained on masked language modelling, which has not been extensively studied.

**Weaknesses:**

1.	The main claim of the paper “sequential learning increasingly higher-order interactions by BERT” is not well supported by the results in Figure 4. The term “sequential learning” implies that (i) at the early stage of training, the model first learns low-order interactions *without learning high-order interactions*, (ii) at the later stage of training, the model learns high-order interactions. However, although Figure 4 can support the above point (ii), but it cannot support point (i). The reason is as follows: at epoch 3, the testing loss on 7-body clones is already significantly lower than the testing loss on 3-body clones, which means that the model already learns part of the 7-order interactions at the early training stage. This further indicates that the model learns 3-order, 5-order, and 7-order interactions at the same time in the early stage of training.

2. Figure 4 is only tested on one instance, thus lacking statistical significance. I encourage the authors show the test loss on more individual instances as well as the average test loss across all instances to consolidate this conclusion.

3.  The paper is hard to follow due to inconsistent and undefined notations.

(1) I suggest adding a preliminary section with a formal definition of what “factored attention” means. There should be at least an equation showing how factored attention operates given a sequence of input tokens, what parameters it contains, etc.

(2) In Line 93, $\mu$ and $\alpha$ are not defined. I guess $\mu$ means the $\mu$-th sample in the training set, but I’m not sure what $\alpha$ refers to.

(3) According to Line 84 “each token $s_i$ is an element taking values in a discrete vocabulary $s_i\in \\{1,...,|\mathbb{V}|\\}$”, the equation $s_{i \alpha}^\mu=1$ in Line 93 means that we are always measuring the probability of generating the first token in the vocabulary. This seems unreasonable.

(4) In Equation (2), $A$ and $V$ are not defined.

(5) The meaning of the subscript $\mu$ in Equation (3) seems to conflict with the meaning of the superscript $\mu$ in Line 93.

(6) In Line 134, $p$ is not defined. Moreover, it is not clear whether the superscript $p$ is an exponent, or simply an index.

4.	Figure 1 is misleading. The input sentence is shown as a folded tape, which highly resembles the structure of amino acid. However, this paper has no connection with amino acid, nor does it discuss about the spatial position of each token. Thus, I think this design is quite redundant. Why not simply write the input sentence on a horizontal line?

**Questions:**

Figure 2(c) shows the mean squared displacement (MSD) of the weight for a 3-layer transformer with *factored attention* during training. It shows that the weight of the last layer first begins to change, then the second layer, and at last the first layer. I wonder if this phenomenon can also be observed on a transformer with *standard attention*.

**Limitations:**

Limitations are properly stated in Section 5.

---

> ### Author Rebuttal · Authors · 2024-08-07
>
> We thank the referee for their detailed questions and their feedback. Below we
> reply point-by-point; we hope that our replies alleviate your concerns. If so, we would appreciate if you revisit your rating; if not, we look forward to discussing further during the discussion period.
>
> > Support for our main claim in Figures 1 and 4:
>
> We're glad the reviewer agrees that our results show that transformers learn
> higher-order interactions between tokens in the later stages of training, point
> (ii). However, we think our results support also point (i) both in experiments
> on real data and in the controlled setting with synthetic data. The left panel of
> Figure 1 shows that the test loss curves for the different clones overlap during
> the initial stage of training. Specifically, even after the initial plateau,
> between 5000 and 6000 training steps, the loss curves on the various clones
> continue to overlap despite the quick improvement in test error.  The crossover
> between the two-body and the higher-order interactions is at step 15000, and
> only at 22000 the models start differing significantly. These results show that,
> at the early stage of training, the model first learns low-order interactions
> _without learning high-order interactions_. Furthermore, point (i) is clearly
> supported by the experiments on the synthetic dataset (Section 2) and from the
> analytical computation (Section 3.1).
>
> > I encourage the authors show the test loss on more individual
> > instances as well as the average test loss across all instances to consolidate
> > this conclusion.
>
> We have performed our experiments using various random seeds, as specified in
> the figure captions, and found the results to be consistent across different
> seeds. We have updated the plots by adding the corresponding error bars; we show
> an example in the rebuttal figure 1, where we redrew Fig 1 with error bars. We
> will update all figures in this way in the revised version.
>
> > Clarifying the nature of “factored attention”
>
> We define factored attention, where the attention matrix depends only on the
> position of the tokens, by writing the corresponding probability for the masked
> token $p_{\mathrm{mlm}}$ in Eq (2). To  clarify this we will first define only
> the factored self-attention layer, and then the equation for
> $p_{\mathrm{mlm}}$. The trainable parameters in a factored self-attention layer
> are the input-independent attention weights $A$ (an $L\times L$ matrix, with $L$
> the length of the input sequence) and the value matrix $V$.
>
> > (2) In Line 93, $\mu$ and $\alpha$ are not defined. I guess $\mu$ means the
> > $\mu$-th sample in the training set, but I’m not sure what $\alpha$ refers to.
>
> You are right: $\mu$ is an index that runs over training examples; $\alpha$
> indexes positions along the sequence. We will clarify this in the revised
> manuscript.
>
> > (3) According to Line 84 “each token is an element taking values in a discrete
> > vocabulary ”, the equation in Line 93 means that we are always measuring the
> > probability of generating the first token in the vocabulary. This seems
> > unreasonable.
>
> We wrote the equation in line 93 for the first token to keep the equation
> readable; this equation (and any other) naturally extends to predicting any
> other token.
>
> > (4) In Equation (2), $A$ and $V$ are not defined.
>
> Thank you for pointing this out -- $A$ and $V$ are the attention and the value
> matrix, respectively. We will add a comment.
>
> > (5) The meaning of the subscript $\mu$ in Equation (3) seems to conflict with the
> > meaning of the superscript in Line 93.
>
> In both equations, $\mu$ is simply a "dummy" index that is summed over.
>
> > (6) In Line 134, $p$ is not defined. Moreover, it is not clear whether the
> > superscript is an exponent, or simply an index.
>
> In this equation, $p$ is also a dummy index. We now see that this might clash
> with the notation for probabilities, and use a different dummy index for this
> sum.
>
> > Regarding the design of Figure 1
>
> We have chosen the folded tape shape because it allows highlighting groups of words which are far away, but (qualitatively) form a group of interactors. Importantly, the factored attention architecture learns exactly the strength of the many-body  interactions along the sequence. This is the message that we would like to convey with the figure. These days we attempted to follow the reviewer’s suggestion and write the sentence on a horizontal line, but we did not manage to find a graphically satisfactory alternative.
>
>
> > Repeating the experiments on synthetic data with a
> > transformer with standard attention.
>
> This is a great question, and inspired us to perform an additional analysis. We
> trained a standard three-layer transformer encoder on clones of a synthetic
> datasets characterized by interactions involving up to four bodies. Being a
> standard transformer, this architecture also had several layers of Layer
> Normalization, which mitigate the occurrence of plateaus even on synthetic
> datasets. However, the plot for this standard transformer in Fig
> 1 of the rebuttal figures clearly shows sequential learning also in this
> case.

---

> > ### Comment · Reviewer_Kh1E · 2024-08-10
> >
> > Thank you for your point-by-point response. I appreciate the new experiment on the transformer with standard attention. Regarding the notations, I encourage the authors to clarify all the mentioned issues in the future manuscript, especially replacing the dummy index $\mu$ in Eq. (3) and $p$ in Line 134 with other appropriate symbols.
> >
> > I have further questions regarding Fig. 4 and Fig. 1. The authors respond to my concern about the insufficient empirical support of Fig.4 for the main claim by referring to Fig.1. However, I find that the results in Fig.1 and Fig.4 seem to conflict with each other. In Fig.1, the test losses on 3-body, 5-body, and 7-body clones are almost the same in the early stage of training (before Step 5000), while in Fig.4, the test losses on 3-body, 5-body, and 7-body clones significantly differ in the early stage of training (epoch 3). Could you provide a further explanation?

---

> > > ### Author Response · Authors · 2024-08-10
> > >
> > > Thank you for engaging with our rebuttal, we appreciate your effort.
> > >
> > > > Regarding the notation
> > >
> > > Thanks again for spotting these typos; we have already changed the indices in those sums for clarity.
> > >
> > > > Further questions regarding Fig. 4 and Fig. 1
> > >
> > > Thank you for following up on this.
> > > - We drew Fig 4 using the same data as Fig 1. In this experiment, 1 epoch = 3000 steps of SGD. Fig 1 shows that the test losses of the transformer on the different clones are different after 9k steps (=3 epochs). Hence the blue dots in Fig 4, which represent the performance of the transformer on the different clones (from left to right) after three epochs, have different values.
> > > - Sequential learning in sense (ii) of your definition (learning higher-order interactions at a later stage of training) is borne out by Fig 4 since the performance on the three-body clone does not improve at later epochs (left-most column), while the performance on the five-body clones continues to improve until ~epoch 10, and improves on the seven-body clone for even longer.
> > > - Sequential learning in sense (i) of your definition (learning low-order interactions without learning high-order interactions) can be seen in Fig 1 during the first two epochs, where the loss curves on the various clones continue to overlap despite the quick improvement in test error.
> > >
> > > Does this answer your questions? We will add epochs to the $x$-axis of Fig 1 for increased readability, and explicitly state the number of steps per epoch for the experiments. Please let us know if there are other changes that could help the reader, or if you have any other questions.

---

> > > > ### Comment · Reviewer_Kh1E · 2024-08-11
> > > >
> > > > Thank you for the clarification. The main issue lies in the inconsistent label of the x-axis in Fig.1 and Fig.4, which causes my confusion. I'm happy to see this being resolved in the future version of the manuscript, and I'm raising my score.

---

### Author Rebuttal · Authors · 2024-08-07

We thank the reviewers for taking the time to review of our manuscript. In our paper, we show that BERT-style transformers trained using masked language modeling learn increasingly complex interactions among input tokens. To conduct this analysis, we develop a procedure to generate clones of a given natural language data set, which rigorously capture the interactions between tokens up to a specified order.

Prompted by the thoughtful questions in the replies, we conducted a number of additional experiments to solidify our points:
- On the left of Fig 1 of the attached PDF, we show sequential learning in the same setting as in Figure 1 of the main submission, but on a different data set: WikiText.
- On the right of the same figure, we show sequential learning of a vanilla transformer with standard attention trained on our synthetic data set.
- We have also added an additional figure showing the statistical uncertainty in the test loss trajectories for Figure 1.

We hope that these results alleviate your concerns. If not, we look forward to discussing further during the discussion period.

---

### Decision · Program_Chairs · 2024-09-25

**Decision:**

Accept (poster)

**Comment:**

This paper studies the simplicity bias in BERT-like models, by asking how transformers learn high-order interactions between tokens. The reviewers all appreciated this work, in particular the importance of the relatively underexplored problem, the novel (theoretical) insights, and the clarity. Main concerns regarding framing and some claims not being supported, overhead, missing related work, and the quality of the results have largely been addressed in the rebuttal.